# GuideCO: Training Objective-Guided Diffusion Solver with Imperfect Data for Combinatorial Optimization

## Abstract

Combinatorial optimization (CO) problems have widespread applications in science and engineering but they present significant computational challenges. Recent advancements in generative models, particularly diffusion models, have shown promise in bypassing traditional optimization solvers by directly generating near-optimal solutions. However, we observe an exponential scaling law between the optimality gap and the amount of training data needed for training diffusion-based solvers. Notably, the performance of existing diffusion solvers relies on both quantity and quality of training data: they perform well with abundant high quality training data labeled by exact or near-optimal solvers, while suffering when high-quality labels are scarce or unavailable. To address the challenge, we propose GuideCO, an objective-guided diffusion solver for combinatorial optimization, which can be trained on imperfectly labelled datasets. GuideCO is a two-stage generate-then-decode framework, featuring an objective-guided diffusion model that is further reinforced by classifier-free guidance for generating high-quality solutions on any given problem instance. Experiments demonstrate the improvements of GuideCO against baselines when trained on imperfect data, in a range of combinatorial optimization benchmark tasks such as TSP (Traveling Salesman Problem) and MIS (Maximum Independent Set).

## 1 Introduction

Combinatorial optimization (CO) problems present fundamental challenges in computational science, as they involve finding optimal solutions from an exponentially large set of possibilities. Traditionally, approaches to solving CO problems have relied integer programming (IP) or carefully crafted heuristics (Gonzalez, 2007; Arora, 1996), requiring substantial computational resources and extensive domain knowledge.

Recently, generative models have emerged as powerful and promising tools for tackling combinatorial optimization problems. Variational Autoencoders (VAEs) (Hottung et al., 2021) and diffusion models (Sun & Yang, 2023) have demonstrated their effectiveness in classic challenges such as the Traveling Salesman Problem (TSP) and Maximal Independent Set (MIS). Graph generators (Li et al., 2023; You et al., 2019) have shown great potential in solving complex problems like Satisfiability (SAT) and Mixed-Integer Linear Programming (MILP). Beyond traditional benchmarks, generative models are now being successfully applied to real-world combinatorial design tasks such as chip design (Du et al., 2024; Cheng et al., 2022) and game design (Cui et al., 2022), highlighting their adaptability to practical applications.

Most notably, recent adaptations of diffusion models (Sun & Yang, 2023) to CO have achieved state-of-the-art performance for solving TSP. The success of diffusion models in CO can be attributed to their supervised progressive denoising paradigm, which can directly model the multi-modal joint distribution over the solution space, and enjoys high simplicity in training process at the same time. Therefore, it avoids the sequential generation bottleneck of autoregressive solvers (Vinyals et al., 2015; Kool et al., 2018) and also surpasses the instability in RL-based methods (Wu et al., 2021; Chen & Tian, 2019).

However, despite these advantages brought by the diffusion modeling paradigm, we observe an exponential scaling law for the relationship between optimality gap and training data quantity (blue curve in Fig. 1). Furthermore, the performance of diffusion solvers rely heavily on the training data quality: to achieve the best performance, training instances are required to be labeled by exact or near-optimal solvers. Without such high-quality labels, their performance significantly declines (green curve in Fig. 1

In response to the challenge we identified, we investigate the following questions in this paper:

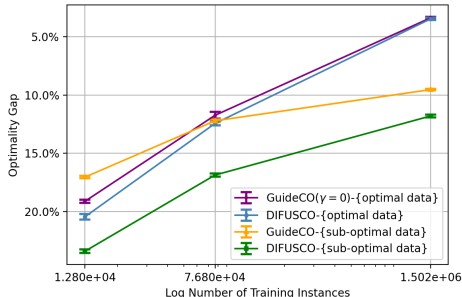

Figure 1: **Exponential data scaling law in diffusion solvers.** Tested on TSP-50 benchmark. The optimality gap sup-optimal data is $9.42\%$.

**Q1: Can we mitigate the performance drop in diffusion solvers when training instances are labeled with sub-optimal solutions?**

**Q2: Can we train diffusion solvers to achieve good solving quality while solely using instances labeled with sub-optimal solutions?**

At their core, these two questions call for the extrapolation ability of diffusion solvers: to learn how to generate better solutions than what have been seen in the training dataset. To this end, we propose GuideCO, an objective-guided training framework for diffusion solvers, which is illustrated in Fig.2. GuideCO is a two-stage generate-then-decode framework, featuring an objective-conditioned diffusion model that is further reinforced by a classifier-free guidance, to generate high-quality solutions even when training with imperfectly labeled instances.

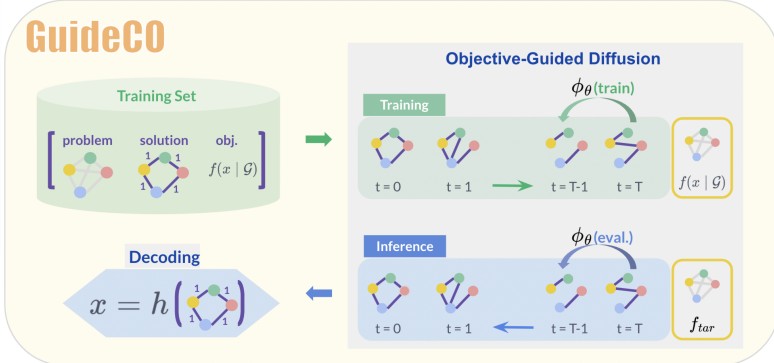

Figure 2: Illustrations of the GuideCO framework with objective-guided diffusion model.

GuideCO offers a two-stage generate-then-decode strategy (§ 3.1): a solution graph is firstly generated via a graph diffusion model on the original CO problem graph, and a final solution is then decoded via greedy methods on top of the solution graph. This two-stage strategy has a intriguing link to a bi-level relaxation of original CO problems, and this design is empirically observed to be beneficial when training with imperfect data. The key advancement in GuideCO is a objective-conditioned diffusion model (§ 3.2), motivated by its ability to differentiate a generation processes under varying conditions and then generate novel data points aligned with the input condition (Ajay et al., 2022; Sohl-Dickstein et al., 2015; Yuan et al., 2024). Therefore, in the optimization context, integrating objective as a condition enables diffusion models to differentiate generation processes with varying levels of optimality. Thus during inference, GuideCO can guide the generation process to a direction with higher optimality. In addition, we propose a novel classifier-free guidance for CO (Ho & Salimans, 2022) that can further reinforce the guidance strength (§3.2.2).

Experiment results demonstrate positive answers to the two question we have raised. We evaluated GuideCO on two benchmark tasks such as TSPs (with varying sizes of 50, 100, 500, 1000) and MIS on SATLIB and weighted/unweighted Erdos–Rényi graphs. For Question 1, GuideCO consistently outperforms DIFUSCO across all benchmarks when both models are trained with instances with sub-optimal labels (Tables in § 4.3 and § 4.4), delivering a decisive positive answer to Q1. For the

more ambitious Question 2, GuideCO demonstrates a strong potential. Despite using sub-optimal data, in TSP50/100, GuideCO outperforms DIFUSCO trained with solver-labeled instances (Table 2, 3); in MIS benchmarks, GuideCO trained with heuristic-labeled data has a matched performance to DIFUSCO trained with solver-labeled data ($-1\%$ to $+0.45\%$ performance gain in Table 5), while collecting heuristic-labeled training instances for GuideCO is over $40\times$ faster (Table 1).

In contrast to a recent line of work that proposes objective-aware methods to improve diffusion solvers (Li et al., 2024a; Yoon et al.) at the post-training stage, our paper studies the training process of diffusion solvers. The highlights of our contributions in this paper are: 1) we identify an exponential data scaling law in training diffusion solvers; 2) we propose an objective-guided diffusion model featuring a classifier-free guidance to generate high-quality solutions from imperfect training data; 3) we conducted extensive experiments to demonstrate that our method outperforms baseline models when training with imperfectly-labeled data.

## 2 PROBLEM SETUP

We start with a formal problem set-up of combinatorial optimization (CO). In § 2.2, we introduce the imperfect training data to use in GuideCO on its generation time and quality.

### 2.1 COMBINATORIAL OPTIMIZATION ON GRAPHS

A lot of combinatorial optimization (CO) problems can be formulated with graphs (Lucas, 2014), so we formally formulate CO problems with graph structure by:

$$\min_{\boldsymbol{x}} \text{ or } \max_{\boldsymbol{x}} f(\boldsymbol{x} \mid \mathcal{G}) \quad \text{s.t.} \quad c_i(\boldsymbol{x}, \mathcal{G}) \leq 0, \text{ for } i = 1 \ldots I. \tag{1}$$

where $\boldsymbol{x}$ denotes the solution, $f(\boldsymbol{x} \mid \mathcal{G})$ denotes the objective function given input graph $\mathcal{G}$ and $c_i(\boldsymbol{x}, \mathcal{G}) \leq 0$ represents the set of constraints. The goal of CO is to find the solution $\boldsymbol{x}$ satisfying (1) for any input graph $\mathcal{G}$, which specifies an instance of the problem. To present our method with higher clarity, three specific CO problems are provided as examples. We start with some necessary notations for defining those problems.

**Notations.** Suppose graph $\mathcal{G}$ is represented by $\mathcal{G} = \langle \boldsymbol{V}, \boldsymbol{E} \rangle$. $\boldsymbol{V} \in \mathbb{R}^{n \times d_v}$ contains all node features, $n$ is the number of nodes in $\mathcal{G}$ and vector $\boldsymbol{v}_i \in \mathbb{R}^{d_v}$ in the $i$-th row of $\boldsymbol{V}$ is the feature for node $i$. $\boldsymbol{E} = \{\boldsymbol{e}_{ij} | \boldsymbol{e}_{ij} \in \mathbb{R}^{d_e}, 1 \leq i, j \leq n\}$ consists of all edge features, $\boldsymbol{e}_{ij}$ is the feature of edge between node $i$ and $j$. For the problems we consider in this paper, solution $\boldsymbol{x}$ can be a permutation or a subset of all nodes in $\mathcal{G}$. For a graph with $n$ nodes, define the set of its node indices as $\mathbb{S} = \{1, 2, \cdots, n\}$. If $\boldsymbol{x}$ is a permutation, then $\boldsymbol{x}$ is defined as a bijection from $\mathbb{S}$ to itself s.t. each node appears once and only once in $\{\boldsymbol{x}(i), i = 1, \cdots, n\}$. If $\boldsymbol{x}$ is a subset of all nodes, then $\boldsymbol{x}$ is directly defined as a subset of $\mathbb{S}$ s.t. $\boldsymbol{x} \subset \mathbb{S}$, we still denote the elements in $\boldsymbol{x}$ as $\boldsymbol{x}(i)$ so that $\boldsymbol{x} = \{\boldsymbol{x}(i), i = 1, \cdots, k\}$ where each node in $\mathcal{G}$ appears at most once and $k \leq n$.

Due to space limit, in what follows, brief formulations of three example CO problems are presented, more detailed and rigorous formulations are deferred to Appendix A.

**Problem 1** (Travelling Salesman Problem **(TSP)**). *Given a graph $\mathcal{G}$ with nodes representing a list of cities and their locations, TSP aims to find the shortest route that visits each city exactly once and returns to the origin. In TSP, node feature $\boldsymbol{v}_i \in \mathbb{R}^2$ is the 2D coordinate of node $i$ and edge feature $\boldsymbol{e}_{ij} \in \mathbb{R}$ is the Euclidean distance between node $i$ and $j$. In (1), the $c_i$'s constraint $\boldsymbol{x}$ to be a permutation and the objective to minimize is $f(\boldsymbol{x} \mid \mathcal{G}) = \sum_{i=1}^{n-1} \boldsymbol{e}_{\boldsymbol{x}(i), \boldsymbol{x}(i+1)} + \boldsymbol{e}_{\boldsymbol{x}(n), \boldsymbol{x}(1)}$.*

**Problem 2** (Maximum Independent Set **(MIS)**). *MIS is to find the largest independent set for any given undirected graph $\mathcal{G}$. In MIS, node feature $\boldsymbol{v}_i \in \mathbb{N}^*$ is an integer recording the weight of node $i$ ($\boldsymbol{v}_i = 1$ for unweighted case). Edges in $\mathcal{G}$ are binary: $\boldsymbol{e}_{ij} = 1$ means node $i$ and $j$ are connected otherwise disconnect. In (1), $c_i$'s constraint that $\boldsymbol{e}_{\boldsymbol{x}(i), \boldsymbol{x}(j)} = 0$ are satisfied for all node pairs in $x$; and $f(\boldsymbol{x} \mid \mathcal{G})$ to maximize in (1) is defined as $\sum_{\boldsymbol{x}(i) \in \boldsymbol{x}} \boldsymbol{v}_{\boldsymbol{x}(i)}$, counting the weighted size of $\boldsymbol{x}$.*

### 2.2 IMPERFECT TRAINING DATA FOR DIFFUSION SOLVER

Different from previous supervised-learning based solver models Nowak et al. (2018); Sun & Yang (2023); Li et al. (2024a) that consume perfectly labelled data consisting of problem-solution pairs as

$\{(\mathcal{G}, \boldsymbol{x}^*)\}$, where $\boldsymbol{x}^*$ is the exact optimal solution of $\mathcal{G}$ produced by solvers, we work with instances labelled with sub-optimal solutions. Table 1 shows a comparison for the labeling time and quality of using exact and heusitic solvers for TSP and MIP problems. It shows that the generating costs of sub-optimal training data with heuristic methods is way more economical than exact solvers.

| Labeller | TSP-100 | | | Weighted MIS-100 | | |
|---|---|---|---|---|---|---|
| | Length↓ | Gap↓ | Time ↓ | Size↑ | Gap ↓ | Time ↓ |
| **Exact Solver** | 7.76 | — | 13.20 min | 135.40 | — | 40.00min (32 workers) |
| **Heuristic** | 8.72 | 12.29 % | 2.10 min | 122.47 | 9.55 % | 0.91min |

Table 1: Comparison of labelling time and quality between exact and heuristic solvers for TSP-100 and MIS-100 tasks. The exact solver for TSP and MIS is `Concorde` and `Gurobi`, respectively; and the heuristic method for TSP and MIS is `farthest_insertion` and Olmi (2024), respectively. Time reported here is for generating a batch of 12800 training samples.

## 3 GUIDECO: OBJECTIVE-GUIDED DIFFUSION SOLVER

A generative perspective (Sun & Yang, 2023; Li et al., 2024a) has been adopted to seeks one (or multiple) optimal solutions $\boldsymbol{x}^*$ given the problem instance instance $\mathcal{G}$. It naturally corresponds to a conditional generation task: to generate $\boldsymbol{x}$ conditioned on $\mathcal{G}$ according to $P(\boldsymbol{x}^* \mid \mathcal{G})$, a conditional distribution of optimal solutions given the problem instance. GuideCO is primarily based on this generative perspective and designs of Sun & Yang (2023), in which a solution is formulated as binary vectors and $P(\boldsymbol{x}^* \mid \mathcal{G})$ is viewed as a graph-to-vector prediction task, and a final solution is decoded based on the logits of prediction.

### 3.1 GENERATE-THEN-DECODE FRAMEWORK

In this paper, we propose a two-stage generation method that merges diffusion and heuristics methods for effectively generating solutions for CO, generalizing the key designs in Sun & Yang (2023). It's called generate-then-decode: in the first stage a **"solution graph"** $\mathcal{G}^{\boldsymbol{x}}$ is generated based on the **"problem graph"** $\mathcal{G}$, and in the second stage a decoding algorithm $h(\cdot)$ is applied to solve $\mathcal{G}^{\boldsymbol{x}}$ so that the final solution is obtained as $\boldsymbol{x} = h(\mathcal{G}^{\boldsymbol{x}})$. The utilization of heuristic decoding methods can reduce requirement of solution quality generated by diffusion model while without compromising quality.

With this two-stage view, the current generative formulation of CO closely links to the following bi-level relaxation of the original problem:

$$\min_{\boldsymbol{x}, \mathcal{G}^{\boldsymbol{x}}} f(\boldsymbol{x} \mid \mathcal{G})$$
$$\text{s.t. } \boldsymbol{x} \in \arg\min_{\boldsymbol{x}'} \left\{ f(\boldsymbol{x}' \mid \mathcal{G}^{\boldsymbol{x}}) : c_i(\boldsymbol{x}', \mathcal{G}^{\boldsymbol{x}}) \leq 0, \text{ for } i = 1 \ldots I \right\}, \quad (2)$$

here $f$ and $\mathcal{G}$ are the same objective and problem instance in (1), and lower level problem in (2) is approximately solved by the decoding algorithm $h(\cdot)$ in the two-stage process.

We use TSP and MIS as two examples to showcase the two-stage process and the link to bi-level relaxation. The solution graph $\mathcal{G}^{\boldsymbol{x}} = \langle \boldsymbol{V}^{\boldsymbol{x}}, \boldsymbol{E}^{\boldsymbol{x}} \rangle$ generated in the first stage and the solution $\boldsymbol{x}$ output in the second stage are defined as follows:

- **TSP:** $\boldsymbol{x}$ is a permutation of nodes in $\mathcal{G}$. In $\mathcal{G}^{\boldsymbol{x}}$, $\boldsymbol{V}^{\boldsymbol{x}} = \boldsymbol{V}$, i.e. the node features stayed unchanged from the problem graph $\mathcal{G}$. In $\boldsymbol{E}^{\boldsymbol{x}}$, the edge between node $i$ and $j$ exits if and only if it is covered in the tour specified by $\boldsymbol{x}$, i.e. $\boldsymbol{e}_{\boldsymbol{x}(n), \boldsymbol{x}(1)} = 1$ and $\boldsymbol{e}_{\boldsymbol{x}(i), \boldsymbol{x}(i+1)} = 1$ for $i = 1 \cdots n - 1$, $\boldsymbol{e}_{i,j} = 0$ for the rest of positions.
- **MIS:** $\boldsymbol{x}$ is a subset of nodes in $\mathcal{G}$. In $\mathcal{G}^{\boldsymbol{x}}$, $\boldsymbol{E}^{\boldsymbol{x}} = \boldsymbol{E}$, i.e. the edge features stayed unchanged from the problem graph $\mathcal{G}$. The node feature $\boldsymbol{V}^{\boldsymbol{x}}$ have $v_i = 1$ if node $i$ is in the solution subset $\boldsymbol{x}$, otherwise $v_i = 0$.

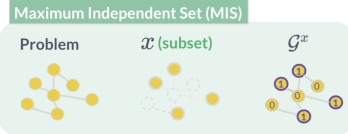

Figure 3: Solutions and solution graphs in TSP and MIS.

In both TSP and MIS, it is easy to see $\boldsymbol{x}$ is a solution of the lower level problem in (2): for TSP the objective is $\min_{\boldsymbol{x}'} -f(\boldsymbol{x}' \mid \mathcal{G}^{\boldsymbol{x}})$ and for MIS that is $\max_{\boldsymbol{x}'} f(\boldsymbol{x}' \mid \mathcal{G}^{\boldsymbol{x}})$, linking the two stage process to a bi-level formulation. As a result, the solution graph $\mathcal{G}^{\boldsymbol{x}}$ reflects a solution $\boldsymbol{x}$, hence $\boldsymbol{x}$ can be obtained through a greedy algorithm based on the distribution of $\mathcal{G}^{\boldsymbol{x}}$ output by the diffusion model. In pracThe specific greedy algorithms (Graikos et al., 2022; Qiu et al., 2022; Sun & Yang, 2023) for TSP and MIS are summarized as follows:

- **TSP:** Sort the logits of $\boldsymbol{E}^{\boldsymbol{x}}$ in the descending order as confidence, sequentially insert edges from high to low confidence if there are no conflicts, until a tour is formed.
- **MIS:** Start with $\boldsymbol{x} = \emptyset$. Inserting nodes into $\boldsymbol{x}$ in the descending order of $\boldsymbol{V}^{\boldsymbol{x}}$'s logits as long as there are no conflicts, until all nodes are gone over.

### 3.2 OBJECTIVE-GUIDED DIFFUSION MODELS

In this section, we present our objective-guided diffusion model featuring objective-conditioned diffusion (§ 3.2.1) and guide-reinforced diffusion (§ 3.2.2), and conclude the section with network architecture of GuideCO (§ 3.2.3).

### 3.2.1 OBJECTIVE-CONDITIONED DIFFUSION SOLVER

To develop a diffusion model that maximally utilizes training data sub-optimal labels, we propose a objective-guided diffusion model, which approximates $P(\mathcal{G}^{\boldsymbol{x}} \mid \mathcal{G}, f(\boldsymbol{x} \mid \mathcal{G}))$, incorporating the objective value $f(\boldsymbol{x} \mid \mathcal{G})$ as a separate condition. The diffusion model for modelling $P(\mathcal{G}^{\boldsymbol{x}} \mid \mathcal{G}, f(\boldsymbol{x} \mid \mathcal{G}))$ follows the discrete diffusion formulation in Sun & Yang (2023) for its forward process. W.L.O.G, we only present the process for generating edges in $\mathcal{G}^{\boldsymbol{x}}$, generating nodes is similar.

Categorical noise is progressively added to $\boldsymbol{E}^{\boldsymbol{x}}$ sampled from $P(\boldsymbol{E}^{\boldsymbol{x}} \mid \mathcal{G})$ by formula (3), generating a sequence of latents $\boldsymbol{E}_0^{\boldsymbol{x}} := \boldsymbol{E}^{\boldsymbol{x}}, \boldsymbol{E}_{1:T}^{\boldsymbol{x}} := \boldsymbol{E}_1^{\boldsymbol{x}}, \boldsymbol{E}_2^{\boldsymbol{x}}, \cdots, \boldsymbol{E}_T^{\boldsymbol{x}}$ s.t.

$$q\left(\boldsymbol{E}_t^{\boldsymbol{x}} \mid \boldsymbol{E}_{t-1}^{\boldsymbol{x}}\right) = \mathrm{Cat}\left(\boldsymbol{E}_t^{\boldsymbol{x}}; p = \boldsymbol{E}_{t-1}^{\boldsymbol{x}} \boldsymbol{Q}_t\right) \quad \text{and} \quad q\left(\boldsymbol{E}_t^{\boldsymbol{x}} \mid \boldsymbol{E}^{\boldsymbol{x}}\right) = \mathrm{Cat}\left(\boldsymbol{E}_t^{\boldsymbol{x}}; p = \boldsymbol{E}^{\boldsymbol{x}} \overline{\boldsymbol{Q}}_t\right), \quad (3)$$

where $\mathrm{Cat}(\cdot, p)$ denotes categorical distribution, $\boldsymbol{Q}_t$'s are transition kernel for categorical variables and $\overline{\boldsymbol{Q}}_t = \boldsymbol{Q}_1 \ldots \boldsymbol{Q}_t$. More mathematical details can be found in Appendix§ B.

The backward denoising process of our model is objective conditioned, which denoises $\boldsymbol{E}_t^{\boldsymbol{x}}$ to generate the preceding variable $\boldsymbol{E}_{t-1}^{\boldsymbol{x}}$, based on 4 inputs: the current state $\boldsymbol{E}_t^{\boldsymbol{x}}$, the problem instance $\mathcal{G}$, the objective $f(\boldsymbol{x} \mid \mathcal{G})$ and the time step $t$. The denoiser is learned by model $\phi_\theta$, aiming to align its prediction to the input solution $\boldsymbol{E}_0^{\boldsymbol{x}}$, thus the loss for training $\phi_\theta$ is:

$$\min_\theta \mathbb{E}_{t \sim Unif((0,T])} \left[\text{cross-entropy}\left(\phi_\theta(\boldsymbol{E}_t^{\boldsymbol{x}}, \mathcal{G}, f(\mathbf{x}), t), \boldsymbol{E}_0^{\boldsymbol{x}}\right)\right]. \quad (4)$$

The architecture of objective-guided denoiser $\phi_\theta$ will be introduced in §3.2.3. During inference, we first set a target objective $f_{tar}$ and start the backward diffusion process by sampling $\boldsymbol{E}_T$ from the uniform distribution. Then iteratively at each time step $t$, denoting the prediction of $\phi_\theta(\boldsymbol{E}_t, \mathcal{G}, f_{tar}, t)$ as $\hat{\boldsymbol{E}}_0$, the one-step predecessor $\boldsymbol{E}_{t-1}$ can be generated from the following posterior distribution:

$$\boldsymbol{E}_{t-1} \sim \mathrm{Cat}\left(\boldsymbol{E}_{t-1}; p = \frac{\hat{\boldsymbol{E}}_0 \boldsymbol{Q}_t^\top \odot \hat{\boldsymbol{E}}_0 \bar{\boldsymbol{Q}}_{t-1}}{\hat{\boldsymbol{E}}_0 \bar{\boldsymbol{Q}}_t \boldsymbol{E}_t^\top}\right). \quad (5)$$

After $T$ iterations, the recovered solution graph $\mathcal{G}_0$ is expected to have the same distribution as $P(\mathcal{G}^{\boldsymbol{x}} \mid \mathcal{G}, f_{tar})$ so that the solution decoded out from $\mathcal{G}_0$ has objective equal to $f_{tar}$, if the diffusion model approximates the ground truth distribution well. A theoretical choice of $f_{tar}$ is the optimal objective for the original problem, $f^* = \min_{\boldsymbol{x}} f(\boldsymbol{x} \mid \mathcal{G})$. In practice, one can take $f_{tar}$ as a model hyper-parameter and grid search for proper $f_{tar}$ with validation set (§ 4.2).

### 3.2.2 Guide-Reinforced Diffusion Solver

The performance of objective conditioning can be further enhanced by guidance mechanism (Ho & Salimans, 2022; Nisonoff et al., 2024). In the sequel, we propose a classifier-free guidance for **categorical** diffusion model with **discrete** state, compatible with our objective-conditioned diffusion model (Alg.1).

The denoiser of objective-conditioned diffusion approximates the distribution $P(\mathcal{G}_0^{\boldsymbol{x}} \mid \mathcal{G}_t^{\boldsymbol{x}}, \mathcal{G}, f^*)$, and Bayesian rule suggests:

$$P(\mathcal{G}_0^{\boldsymbol{x}} \mid \mathcal{G}_t^{\boldsymbol{x}}, \mathcal{G}, f^*) \propto P(\mathcal{G}_0^{\boldsymbol{x}} \mid \mathcal{G}_t^{\boldsymbol{x}}, \mathcal{G}) \cdot P(f^* \mid \mathcal{G}_t^{\boldsymbol{x}}, \mathcal{G}), \tag{6}$$

where $P(\mathcal{G}_0^{\boldsymbol{x}} \mid \mathcal{G}_t^{\boldsymbol{x}}, \mathcal{G})$ on the RHS is the unconditioned probability to denoise $\mathcal{G}_t^{\boldsymbol{x}}$. This property suggests a way to further enhance the guidance of objective by denosing with the following probability at each step:

$$\frac{1}{Z} \cdot P(\mathcal{G}_0^{\boldsymbol{x}} \mid \mathcal{G}_t^{\boldsymbol{x}}, \mathcal{G}, f^*) \left( \frac{P(\mathcal{G}_0^{\boldsymbol{x}} \mid \mathcal{G}_t^{\boldsymbol{x}}, \mathcal{G}, f^*)}{P(\mathcal{G}_0^{\boldsymbol{x}} \mid \mathcal{G}_t^{\boldsymbol{x}}, \mathcal{G})} \right)^{\gamma}. \tag{7}$$

In (7), $Z$ is a normalizing factor and $\gamma \geq 0$ controls the strength of guidance. Detailed derivation for (7) is provided in § B.2. To facilitate the classifier-free guidance as (7), we jointly train an unconditioned denoiser to approximate $P(\mathcal{G}_{t-1}^{\boldsymbol{x}} \mid \mathcal{G}_t^{\boldsymbol{x}}, \mathcal{G})$ together with the original conditioned one. Therefore, for training an objective-directed diffusion model with classifier-free guidance, we modify the previous loss in (4) to be

$$\min_{\theta} \mathbb{E}_{t \sim Unif((0,T]), s \sim Bernoulii(p)} \left[ \mathrm{I}\{s = 0\} \cdot \text{cross-entropy} \left( \phi_{\theta}(\mathcal{G}_t^{\boldsymbol{x}}, \mathcal{G}, f(\mathbf{x}), t), \mathcal{G}_0^{\boldsymbol{x}} \right) \right.$$
$$\left. + \mathrm{I}\{s = 1\} \cdot \text{cross-entropy} \left( \phi_{\theta}(\mathcal{G}_t^{\boldsymbol{x}}, \mathcal{G}, \emptyset, t), \mathcal{G}_0^{\boldsymbol{x}} \right) \right], \tag{8}$$

here $s$ is a random seed for determining which samples are held out for training unconditioned denoiser, and an empirical choice of $p$ is 0.1. It is worth mentioning that both conditioned and unconditioned training share the same model $\phi_{\theta}$: if a sample is sent to unconditioned training, then it will go through the forward pass of $\phi_{\theta}$ with the objective condition being masked. The pseudo-code of training and inference in objective-guided diffusion model is provided in Algorithm 1.

---

**Algorithm 1** `Training`

1: **Input**: Training dataset: $\mathcal{D} = \{(\mathcal{G}, \boldsymbol{x}, f(\boldsymbol{x}))\}$
2: **Initialize**: denoising network $\phi_{\theta}(\cdot)$,
   mask probability $p$,
   $T \leftarrow$ total diffusion steps,
   $\{\boldsymbol{Q}_t\}, \{\overline{\boldsymbol{Q}}_t\} \leftarrow$ noise transitions.
3: **Pre-process the Training Data**: represent $\boldsymbol{x}$ as solution graph $\mathcal{G}^{\boldsymbol{x}} = \langle \boldsymbol{V}^{\boldsymbol{x}}, \boldsymbol{E}^{\boldsymbol{x}} \rangle$ as in § 3.1, and obtain $\mathcal{D} = \{(\mathcal{G}, \mathcal{G}^{\boldsymbol{x}}, f(\boldsymbol{x}))\}$.
4: **for** each training step **do**
5:   Sample $t$ from $[0, T]$.
6:   Sample $s \sim Ber(p)$.
7:   **Add Noise:**

$$q\left(\boldsymbol{E}_t^{\boldsymbol{x}} \mid \boldsymbol{E}^{\boldsymbol{x}}\right) = \text{Cat}\left(\boldsymbol{E}_t^{\boldsymbol{x}}; p = \boldsymbol{E}^{\boldsymbol{x}}\overline{\boldsymbol{Q}}_t\right).$$

8:   **Update** $\theta$ with one gradient step w.r.t loss (8).
9: **end for**

---

**Algorithm 1** `Inference`

1: **Input**: denoising network $\phi_{\theta}(\cdot)$, problem $\mathcal{G} = \langle \boldsymbol{V}, \boldsymbol{E} \rangle$, target objective $f_{tar}$, guidance strength $\gamma$, decoding algorithm $h(\cdot)$.
2: **Initialize**: $T \leftarrow$ total diffusion steps
   $N \leftarrow$ # of nodes in $\mathcal{G}$,
   $\{\boldsymbol{Q}_t\}, \{\overline{\boldsymbol{Q}}_t\} \leftarrow$ noise transitions,
3: **Sample:** $\boldsymbol{E}_T \leftarrow \{e_{i,j}\}_{N \times N}, e_{i,j} \sim Ber(\frac{1}{2})$.
4: **for** $t = T, T - 1, \cdots, 1$ **do**
5:   **Denoising Step:**

$$\hat{\boldsymbol{E}}_0 \sim \phi_{\theta}(\mathcal{G}, \boldsymbol{E}_t, t, f_{tar}) \left( \frac{\phi_{\theta}(\mathcal{G}, \boldsymbol{E}_t, t, f_{tar})}{\phi_{\theta}(\mathcal{G}, \boldsymbol{E}_t, t, \emptyset)} \right)^{\gamma},$$

   sample $\boldsymbol{E}_{t-1}$ with (5).

6:   $t \leftarrow t - 1$.
7: **end for**
8: **Decode**: $\boldsymbol{x} \leftarrow h(\mathcal{G}^{\boldsymbol{x}}), \mathcal{G}^{\boldsymbol{x}} = \langle \boldsymbol{V}, \boldsymbol{E}_0 \rangle$.

---

### 3.2.3 Objective-Guided Denoising Network

In (8), our objective-directed denoiser $\phi_{\theta}(\mathcal{G}_t^{\boldsymbol{x}}, \mathcal{G}, f(\mathbf{x}), t)$ takes as input the noisy solution graph $\mathcal{G}_t^{\boldsymbol{x}}$, the problem graph $\mathcal{G}$, the objective value of solution $f(\mathbf{x})$, the time step $t$, to predict the clean solution graph $\mathcal{G}_0^{\boldsymbol{x}}$. Since the denoiser should support predict both node and edge features in the solution graph, we adopt an anisotropic graph neural network (Joshi et al., 2019; Qiu et al., 2022; Sun & Yang, 2023) as the backbone, which can produce embeddings for both nodes and edges.

To incorporate and process the objective information, we propose the following architecture for objective-directed graph denoiser, which is also illustrated in Fig.4.

**Objective-Aware Graph-Based Denoiser.** Let $h_i^\ell$ and $m_{ij}^\ell$ denote the node and edge features at layer $\ell$ for node $i$ and edge $ij$. To process timestep $t$ and objective $f(x)$, we adopt the positional encoding (Vaswani, 2017) and denote: $t = \text{pos}(t)$ and $f = \text{pos}(f(x))$. The features at the next layer is propagated with an anisotropic message passing scheme, engaging the positional encodings of both timestep $t$ and objective $f(x)$:

$$\hat{m}_{ij}^{\ell+1} = P^\ell m_{ij}^\ell + Q^\ell h_i^\ell + R^\ell h_j^\ell,$$

$$m_{ij}^{\ell+1} = m_{ij}^\ell + \text{MLP}_m(\text{BN}(\hat{m}_{ij}^{\ell+1})) + \text{MLP}_t(t) + \text{MLP}_f(f),$$

$$h_i^{\ell+1} = h_i^\ell + \alpha(\text{BN}(U^\ell h_i^\ell + \mathcal{A}_{j\in\mathcal{N}_i}(\sigma(\hat{m}_{ij}^{\ell+1}) \odot V^\ell h_j^\ell))),$$

where in layer $\ell$, $U^\ell, V^\ell, P^\ell, Q^\ell, R^\ell \in \mathbb{R}^{d\times d}$ and $\text{MLP}(\cdot)$ are learnable. $\text{MLP}(\cdot)$ with subscripts $m, t, f$ all denote a 2-layer multilayer perceptron. $\alpha$, BN, $\mathcal{A}$, $\sigma$ denote the ReLU (Krizhevsky et al., 2010) activation, batch normalization (Ioffe, 2015), aggregation function SUM pooling (Xu et al., 2018) and sigmoid function, respectively. $\odot$ is the Hadamard product, $\mathcal{N}_i$ denotes the neighborhoods of node $i$. We use 12 layers with hidden dimension $d = 256$ following Sun & Yang (2023). Lastly, a Sigmoid activation is applied to the final layer embeddings of nodes or edges, which is then to predict a binary cross entropy loss between candidate solution graph vs. input solution graph.

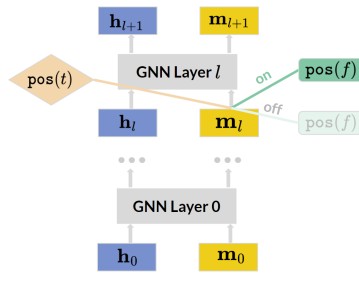

Figure 4: Architecture of objective-directed denoiser.

**Double Graph Conditioning.** It is worth noticing that our denoiser $\phi_\theta(\mathcal{G}_t^x, \mathcal{G}, f(\mathbf{x}), t)$ takes two graphs as input: the problem graph $\mathcal{G}$ and the noisy solution graph $\mathcal{G}_t^x$. To pass both graphs into the anisotropic GNN above, we concatenate the positional encoding of node/edge features in both graphs: suppose $\mathcal{G} = \langle V, E \rangle$ and $\mathcal{G}_t^x = \langle V_t^x, E_t^x \rangle$, we pass $h_i^0 = (\text{pos}(V_{(i)}), \text{pos}(V_{t,(i)}^x))$ and $m_{ij}^0 = (\text{pos}(E_{(i,j)}), \text{pos}(E_{t,(i,j)}^x))$ into the GNN layers. More implementation details for concatenating input graphs in specific problems are provided in Appendix C.

## 4 EXPERIMENTS

### 4.1 EXPERIMENTAL SETUP

Please note experiments are conducted for scenarios where training data is labelled by heuristics but not exact solvers. The setup for data collection, including the choice of heuristics, and baselines for comparison are specified in separate subsections for each problem. Hyper-parameters are as follows.

**Hyper-parameters in diffusion models.** Following DIFUSCO (Sun & Yang, 2023), the models for all problems are trained with 1000 denoising steps, i.e., $T = 1000$, while during inference, we adopt 50 inference steps. We provide in § 4.2 an ablation study on the choices of other two important hyper-parameters for inference: the guidance strength and the target objective value.

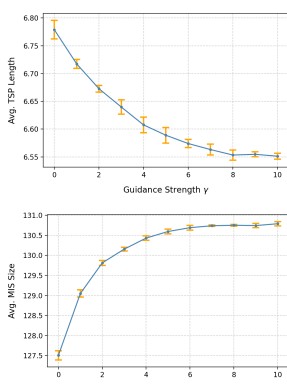

Figure 5: **Effectiveness of guidance strength** on TSP-50 and Weight MIS-100.

### 4.2 ABLATION STUDY

**Guidance Strength.** We investigate the effectiveness of guidance strength $\gamma$ in (7). To efficiently evaluate this choice, we utilize the TSP-50 and Weighted MIS-100 benchmarks and train both models with 12800 instances. We evaluate the model performance on test set, varying the guidance strength by setting $\gamma \in \{0, 1, 2, \cdots, 10\}$

and taking denoising step with guidance as (7) specifies; evaluation results across 5 random seeds are plotted in Fig.5. When $\gamma = 0$, the model degrades to the basic objective-conditioned diffusion model (Alg. 1), it can be seen from Fig.5 that enlarging guidance $\gamma$ improves the baseline performance reported at $\gamma = 0$. It demonstrates that guidance effectively enhances the model performance, guiding the denoising process towards high-objective value region in the solution space.

**Target Objective Value.** We also evaluate the impact of target objective value $f_{tar}$ at inference time. Fig. 6 is a heat-map on the average objective of generated solutions when jointly varying target $f_{tar}$ and strength $\gamma$. Recall from § 3.2, a theoretical choice of $f_{tar}$ is the optimal objective value $f^*$. While empirically, we observe that good choices of $f_{tar}$ live in a wild range above $f^*$ (for maximization) or below $f^*$ (for minimization), see Fig. 6. Here we provide an reference practice for choosing $f_{tar}$: replace $f^*$ with the average optimal objective value in a small validation set and search the multiplication factor from $[1.1, \cdots, 1.5]$ for maximization or $[0.5, 0.6, \cdots, 0.9]$ for minimization and fix the $f_{tar}$ for inference. The best choice of $f_{tar}$ and $\gamma$ is model specific, one can jointly search the two and freely decide how many values to search. $f_{tar}$ and $\gamma$ will be fixed during inference. All experiment results for GuideCO with guidance enabled is reported under fixed $f_{tar}$ and $\gamma$ for all testing instances.

### 4.3 EXPERIMENTS ON TSP

**Datasets.** Following the standard procedure adopted by Kool et al. (2018); Joshi et al. (2019), training and testing instances of TSP are generated by uniformly sampling $n$ nodes from the unit square $[0, 1]^2$. In the training dataset, instances are labelled by heuristic Farthest_Insertion. We experiment on various problem scales including TSP-50, TSP-100, TSP-500 and TSP-1000.

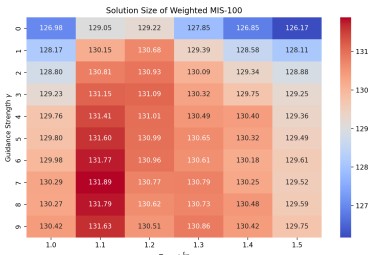

Figure 6: **Joint effectiveness of target objective and guidance strength.** Tested on Weight MIS-100 benchmark.

For TSP-50 and TSP-100, Sun & Yang (2023) uses a total of 1502000 training samples. To measure the data scaling law in diffusion solver, we conduct training for TSP-50 and TSP-100 with 3 sizes of training data: 12800, 76800 and 1502000. For TSP-500/1000, we follow the same number of training instances. The test set for TSP-50/100 is taken from Kool et al. (2018); Joshi et al. (2020) with 1280 instances, and the test set for TSP-500/1000 is with 128 instances for the fair comparison. More details for data and training can be found in Appendix.

**Metrics.** To evaluate model performance, we measure these metrics for TSP: **(i) Length:** the average length of the solution tour, i.e. the objective of solutions in TSP; **(ii) Gap:** the average suboptimality gap of solutions w.r.t. the optimal/near-optimal solution given by the best solver. To compute the optimal objective for TSP, we adopt two solvers: the exact solver Concorde(Applegate et al., 2006) (for TSP-50/100) and the heuristic solver LKH-3(Helsgaun, 2017) (for TSP-500).

**Baselines.** We compare our method to DIFUSCO (Sun & Yang, 2023) when being trained on the same dataset, using the same greedy decoding mechanism, and without 2-opt (Lin & Kernighan, 1973) refinement. This setup directly contrasts the effect of diffusion modelling between the two methods. We include two other baselines: (i) exact solver and (ii) DIFUSCO trained with the same number of instances labelled by solver. The former is for measuring the suboptimality gap of generated solutions, the later is to meaure the performance drop of DIFUSCO when trained with low-qualitiy data labelled by heuristics.

**Results and Analysis** Experiment results for TSP-50 is summarized in Table 2 and for TSP-100 is in Table 3 and Table 4 records results for large scale problems TSP-500/1000. GuideCO outperforms DIFUSCO on all sizes, when both models are trained with heuristic-labeled instances. Notably, despite using heuristic-labelled data, in TSP50, GuideCO outperforms DIFUSCO trained with solver-labeled instances, when the number of training instances is 12800 and 76800. Fig. 7 compares the performance to data scaling curve in GuideCO and DIFUSCO: when both are

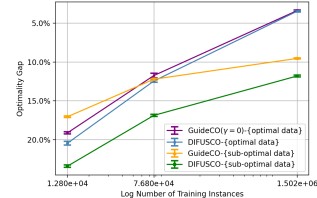

Figure 7: Recall Fig.1. Compare GuideCO and DIFUSCO under varying train size.

| Method | Data Label | TSP-50 (12800) | | TSP-50 (76800) | | TSP-50 (1502000) | |
|---|---|---|---|---|---|---|---|
| | | Length↓ | Gap↓ | Length↓ | Gap↓ | Length↓ | Gap↓ |
| **Concorde** | — | 5.60 | — | 5.60 | — | 5.60 | — |
| DIFUSCO | Solver | 6.74 | 20.36 % | 6.29 | 12.32 % | 5.79 | 3.39 % |
| GuideCO ($\gamma = 0$) | Solver | **6.67** | **19.10** % | **6.25** | **11.60** % | **5.79** | **3.39 %** |
| DIFUSCO | Heuristic | 6.91 | 23.39 % | 6.54 | 16.79 % | 6.26 | 11.79 % |
| GuideCO ($\gamma = 0$) | Heuristic | 6.78 | 21.07 % | 6.38 | 13.93 % | 6.15 | 9.82 % |
| GuideCO | Heuristic | **6.55** | **16.96 %** | **6.28** | **12.14 %** | **6.11** | **9.11 %** |

Table 2: **Results on TSP-50.** $\gamma = 0$ corresponds to the basic objective-conditioned model with no guidance. $\gamma$ is set to $10, 4, 1$ in the last row for the three training sizes.

trained with optimal data, GuideCO (purple) and slightly outperforms DIFUSCO (blue), we test this case with $\gamma = 0$, verifying GuideCO will retain the good performance when trained with optimal data; when trained with sub-optimal data, GuideCO outperforms DIFUSCO trained with either optimal or sub-optimal data in the data-scarce regime between $12800 \sim 76800$. The same data scaling behaviour is observed in TSP-100. For TSP-500/1000, GuideCO mitigates the performance drop in DIFUSCO.

| Method | Data Label | TSP-100 (12800) | | TSP-100 (76800) | | TSP-100 (1502000) | |
|---|---|---|---|---|---|---|---|
| | | Length↓ | Gap↓ | Length↓ | Gap↓ | Length↓ | Gap↓ |
| **Concorde** | — | 7.68 | — | 7.68 | — | 7.68 | — |
| DIFUSCO | Solver | 9.32 | 21.35 % | 8.87 | 15.49 % | 7.95 | 3.52 % |
| DIFUSCO | Heuristic | 9.86 | 28.39 % | 9.47 | 23.31 % | 8.88 | 15.63 % |
| GuideCO ($\gamma = 0$) | Heuristic | 9.69 | 26.17 % | 9.32 | 21.35 % | 8.86 | 15.36 % |
| GuideCO | Heuristic | **9.30** | **21.09 %** | **9.08** | **18.23 %** | **8.83** | **14.97 %** |

Table 3: **Results on TSP-100.** Guidance strength is set to $4, 4, 3$ in the last row.

| Method | Data Label | TSP-500 | | TSP-1000 | |
|---|---|---|---|---|---|
| | | Length↓ | Gap↓ | Length↓ | Gap↓ |
| **LKH-3** | — | 16.54 | — | 23.18 | — |
| DIFUSCO | Solver | 18.47 | 11.67 % | 27.44 | 18.38 % |
| DIFUSCO | Heuristic | 21.60 | 30.59 % | 32.46 | 40.03 % |
| GuideCO | Heuristic | **20.73** | **25.33 %** | **31.82** | **37.27 %** |

Table 4: **GuideCO improves the mitigates the performance drop of DIFUSCO.** Results on TSP-500/1000.

## 4.4 Experiments on MIS

**Datasets.** Three datasets: Weighted MIS-100, SATLIB (Hoos & Stützle, 2000) and unweighted Erdos–Rényi (ER) graphs (Erdos et al., 1960) ($700 \sim 800$ nodes) are tested for the MIS problem. Since Sun & Yang (2023) was only applied to unweighted MIS problems, we add a new weighted MIS-100 dataset consisting of ER graphs with 100 nodes and pairwise connection probability $0.15$, where each node has its weight sampled from $\mathcal{N}(\mu = 5, \sigma = 2)$ and rounded to the nearest integer, we randomly sample $12800/128/1280$ graphs as train/validation/test splits. SATLIB[1] and ER[700-800] (pairwise connection probability $0.15$) are for large scale experiments. We use the same number of training data and the same test instances as in Qiu et al. (2022); Sun & Yang (2023); Li et al. (2024a). The heuristic we choose for labelling training instances is the polynomial algorithm findMIS from Olmi (2024). For Weighted MIS-100 and SATLIB, we also include experiments on mixed dataset: 20% Gurobi-labeled data and 80% findMIS-labeled data.

**Metrics.** Similar to TSP task, we adopt the following metrics to measure model performance for MIS: **(i) Size:** the average size of the solutions, i.e. the objective of solutions in corresponding instances. **(ii) Gap:** the average suboptimality gap of solutions w.r.t. the optimal/near-optimal solution given by the best solver. Solvers for MIS we consider are Gurobi[2] and Kamis[3].

**Results and Analysis.** Experiment results for MIS are summarized in Table 5. GuideCO outperforms DIFUSCO on all datasets, and its performance is significantly improved when optimal data is mixed into the dataset. In contrast, DIFUSCO experiences performance drop, due to its inefficiency in incorporating sub-optimal data values in the model. Remarkably, in Weighted MIS-100 and ER[700-800], GuideCO outperforms findMIS, the algorithm for labelling its training data, by 6% and 3%. It demonstrates the extrapolation ability of GuideCO and its underlying objective-guided diffusion model. Notably, in ER[700-800], GuideCO trained with the heuristic dataset surpasses

---

[1] https://www.cs.ubc.ca/~hoos/SATLIB/Benchmarks/SAT/CBS/descr_CBS.html
[2] https://www.gurobi.com/
[3] https://github.com/KarlsruheMIS/KaMIS

| Method | Data Label | Weighted MIS-100 | | | SATLIB | | | ER[700-800] | | |
|---|---|---|---|---|---|---|---|---|---|---|
| | | Size↑ | Gap↓ | Time↓ | Size↑ | Gap↓ | Time↓ | Size↑ | Gap↓ | Time↓ |
| `Kamis` | — | — | — | — | — | — | — | 44.73 | — | 52.13m |
| `Gurobi` | — | 135.40 | — | 5.00 m | 425.45 | — | 26.00 m | — | — | — |
| `findMIS` | — | 122.37 | 9.62 % | 0.10 m | 421.50 | 0.93 % | 1.45 m | 39.47 | 11.76 % | 2.40 m |
| DIFUSCO | Solver | 133.60 | 1.33 % | 1.30 m | 423.09 | 0.55 % | 0.66 m | 40.93 | 8.51 % | 2.72 m |
| DIFUSCO | Heuristic | 121.78 | 10.06 % | 1.30 m | 420.46 | 1.17 % | 0.66 m | 40.77 | 8.85 % | 2.67 m |
| GuideCO | Heuristic | **130.66** | **3.63 %** | 2.73 m | **420.91** | **1.07 %** | 1.13 m | **41.13** | **8.04 %** | 6.11 m |
| DIFUSCO | Mixed (20% Solver) | 123.80 | 8.57 % | 1.30 m | 420.91 | 1.07 % | 0.66m | — | — | — |
| GuideCO | Mixed (20% Solver) | **132.38** | **2.23 %** | 2.73 m | **421.81** | **0.86 %** | 1.13 m | — | — | — |

Table 5: **Results on Weighted MIS-100, SATLIB, and ER[700-800].** The guidance strength is set to 10, 0.0001 and 8, respectively. We also report the time consumption of testing of all methods.

DIFUSCO trained on the solver dataset, lifting the need of labeling data with solver in this case. For both SATLIB and ER, with classifier-free guidance enabled, GuideCO spends roughly twice the time as DIFUSCO. Taking account for the inference time difference, we evaluate DIFUSCO by taking the maximum performance of two independently sampled solutions for each test instances, the results are 420.54 for SATLIB trained with heuristic-labeled data, 420.98 for SATLIB trained with mixed data, and 40.80 for ER700-800, all being outperformed by GuideCO. In addition, observing that DIFUSCO baseline trained on heuristic data also improves `findMIS` in ER[700-800], suggesting the benefit of the generate-then-decode strategy in the presence of imperfect data.

## 5 RELATED WORK

**Machine Learning for Combinatorial Optimization(ML4CO).** ML4CO has been as a significant area of research over the past decade: previous methods are can be catogorized into autoregressive solver models (Vinyals et al., 2015; Bello et al., 2016; Kool et al., 2018), non-autoregressive solver models (Joshi et al., 2019; Qiu et al., 2022) and reinforcement learning-based improvement heuristics (Wu et al., 2021; Chen & Tian, 2019). Recently, diffusion model has demonstrated its potential in solving CO and DIFUSCO (Sun & Yang, 2023) has achieved the state-of-the-art performance when applied for solving TSP. Li et al. (2024a) and Yoon et al. are two recent works also trying to improve DIFUSCO from a perspective of making the backward generation process in diffusion solver objective-aware. However, in contrast to GuideCO focusing on improving the "pre-training" stage of diffusion solver, their methods take the pre-trained model as a starting point, and make improvement in the "post-training" stage by searching and fine-tuning.

**Optimization Powered by Diffusion Models.** In addition to the recent progress in applying generative models to CO reviewed in the paragraph above, we want to cover some representative works on **"reward-improving diffusion models"**, the reward therein is a direct analogy to the objective in optimization context. A line of works propose to train a reward-conditioned diffusion model, for generating samples with higher rewards at inference time. This paradigm has demonstrated superior performance in black-box optimization (Krishnamoorthy et al., 2023; Li et al., 2024b) and trajectory optimization in reinforcement learning (Ajay et al., 2022). More generally, to improve diffusion models for generating sample quality of high quality measured by an external reward (which could be a white-box, black-box or first-order oracle), guidance methods including classifier-free guidance (Ho & Salimans, 2022) and variants of classifier guidance (Chung et al., 2022; Guo et al., 2024; Bansal et al., 2023), as well as fine-tuning (Clark et al., 2023) methods are good candidates. In this paper, we adopt a discrete version of classifier-free guidance. Nisonoff et al. (2024) proposes a similar "predition-free guidance" for categorical data but with continuous time steps.

## 6 CONCLUSIONS

In this paper, we identified an exponential data scaling law in training diffusion solvers for CO, and their performance highly depends on data quality. To address this challenge, we proposed GuideCO, an objective-guided training framework of diffusion solvers. GuideCO is based on a two-stage generate-then-decode strategy, featuring an objective-guided diffusion model that is further reinforced by classifier-free guidance to better utilize imperfect training instances labelled by polynomial heuristics. Experimental results showed that GuideCO outperformed the baseline DIFUSCO when trained with heuristic-labeled data, and notably GuideCO outperformed DIFUSCO trained with solver-labeled instances when abundant number of training instances is not accessible.

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
