\left(\boldsymbol{x}' \mid \mathcal{G}^{\boldsymbol{x}}\right) : c_i\left(\boldsymbol{x}', \mathcal{G}^{\boldsymbol{x}}\right) \leq 0, \text{ for } i = 1 \ldots I\right\}, \tag{2}$$

here $f$ and $\mathcal{G}$ are the same objective and problem instance in (1), and lower level problem in (2) is approximately solved by the decoding algorithm $h(\cdot)$ in the two-stage process.

We use TSP and MIS as two examples to showcase the two-stage process and the link to bi-level relaxation. The solution graph $\mathcal{G}^{\boldsymbol{x}} = \langle \boldsymbol{V}^{\boldsymbol{x}}, \boldsymbol{E}^{\boldsymbol{x}} \rangle$ generated in the first stage and the solution $\boldsymbol{x}$ output in the second stage are defined as follows:

- **TSP:** $\boldsymbol{x}$ is a permutation of nodes in $\mathcal{G}$. In $\mathcal{G}^{\boldsymbol{x}}$, $\boldsymbol{V}^{\boldsymbol{x}} = \boldsymbol{V}$, i.e. the node features stayed unchanged from the problem graph $\mathcal{G}$. In $\boldsymbol{E}^{\boldsymbol{x}}$, the edge between node $i$ and $j$ exits if and only if it is covered in the tour specified by $\boldsymbol{x}$, i.e. $\boldsymbol{e}_{\boldsymbol{x}(n), \boldsymbol{x}(1)} = 1$ and $\boldsymbol{e}_{\boldsymbol{x}(i), \boldsymbol{x}(i+1)} = 1$ for $i = 1 \cdots n - 1$, $\boldsymbol{e}_{i,j} = 0$ for the rest of positions.
- **MIS:** $\boldsymbol{x}$ is a subset of nodes in $\mathcal{G}$. In $\mathcal{G}^{\boldsymbol{x}}$, $\boldsymbol{E}^{\boldsymbol{x}} = \boldsymbol{E}$, i.e. the edge features stayed unchanged from the problem graph $\mathcal{G}$. The node feature $\boldsymbol{V}^{\boldsymbol{x}}$ have $v_i = 1$ if node $i$ is in the solution subset $\boldsymbol{x}$, otherwise $v_i = 0$.

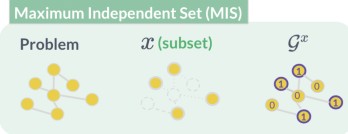

Figure 3: Solutions and solution graphs in TSP and MIS.

In both TSP and MIS, it is easy to see $x$ is a solution of the lower level problem in (2): for TSP the objective is $\min_{x'} -f(x' \mid \mathcal{G}^x)$ and for MIS that is $\max_{x'} f(x' \mid \mathcal{G}^x)$, linking the two stage process to a bi-level formulation. As a result, the solution graph $\mathcal{G}^x$ reflects a solution $x$, hence $x$ can be obtained through a greedy algorithm based on the distribution of $\mathcal{G}^x$ output by the diffusion model. In pracThe specific greedy algorithms (Graikos et al., 2022; Qiu et al., 2022; Sun & Yang, 2023) for TSP and MIS are summarized as follows:

- **TSP:** Sort the logits of $E^x$ in the descending order as confidence, sequentially insert edges from high to low confidence if there are no conflicts, until a tour is formed.
- **MIS:** Start with $x = \emptyset$. Inserting nodes into $x$ in the descending order of $V^x$'s logits as long as there are no conflicts, until all nodes are gone over.

### 3.2 OBJECTIVE-GUIDED DIFFUSION MODELS

In this section, we present our objective-guided diffusion model featuring objective-conditioned diffusion (§ 3.2.1) and guide-reinforced diffusion (§ 3.2.2), and conclude the section with network architecture of GuideCO (§ 3.2.3).

### 3.2.1 OBJECTIVE-CONDITIONED DIFFUSION SOLVER

To develop a diffusion model that maximally utilizes training data sub-optimal labels, we propose a objective-guided diffusion model, which approximates $P(\mathcal{G}^x \mid \mathcal{G}, f(x \mid \mathcal{G}))$, incorporating the objective value $f(x \mid \mathcal{G})$ as a separate condition. The diffusion model for modelling $P(\mathcal{G}^x \mid \mathcal{G}, f(x \mid \mathcal{G}))$ follows the discrete diffusion formulation in Sun & Yang (2023) for its forward process. W.L.O.G, we only present the process for generating edges in $\mathcal{G}^x$, generating nodes is similar.

Categorical noise is progressively added to $E^x$ sampled from $P(E^x \mid \mathcal{G})$ by formula (3), generating a sequence of latents $E_0^x := E^x, E_{1:T}^x := E_1^x, E_2^x, \cdots, E_T^x$ s.t.

$$q\left(E_t^x \mid E_{t-1}^x\right) = \mathrm{Cat}\left(E_t^x; p = E_{t-1}^x Q_t\right) \quad \text{and} \quad q\left(E_t^x \mid E^x\right) = \mathrm{Cat}\left(E_t^x; p = E^x \overline{Q}_t\right), \quad (3)$$

where $\mathrm{Cat}(\cdot, p)$ denotes categorical distribution, $Q_t$'s are transition kernel for categorical variables and $\overline{Q}_t = Q_1 \ldots Q_t$. More mathematical details can be found in Appendix§ B.

The backward denoising process of our model is objective conditioned, which denoises $E_t^x$ to generate the preceding variable $E_{t-1}^x$, based on 4 inputs: the current state $E_t^x$, the problem instance $\mathcal{G}$, the objective $f(x \mid \mathcal{G})$ and the time step $t$. The denoiser is learned by model $\phi_\theta$, aiming to align its prediction to the input solution $E_0^x$, thus the loss for training $\phi_\theta$ is:

$$\min_\theta \mathbb{E}_{t \sim Unif((0,T])} \left[\text{cross-entropy}\left(\phi_\theta(E_t^x, \mathcal{G}, f(\mathbf{x}), t), E_0^x\right)\right]. \quad (4)$$

The architecture of objective-guided denoiser $\phi_\theta$ will be introduced in §3.2.3. During inference, we first set a target objective $f_{tar}$ and start the backward diffusion process by sampling $E_T$ from the uniform distribution. Then iteratively at each time step $t$, denoting the prediction of $\phi_\theta(E_t, \mathcal{G}, f_{tar}, t)$ as $\hat{E}_0$, the one-step predecessor $E_{t-1}$ can be generated from the following posterior distribution:

$$E_{t-1} \sim \mathrm{Cat}\left(E_{t-1}; p = \frac{\hat{E}_0 Q_t^\top \odot \hat{E}_0 \bar{Q}_{t-1}}{\hat{E}_0 \bar{Q}_t

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

{\boldsymbol{m}}_{ij}^{\ell+1} = P^{\ell}\boldsymbol{m}_{ij}^{\ell} + Q^{\ell}\boldsymbol{h}_i^{\ell} + R^{\ell}\boldsymbol{h}_j^{\ell},$$

$$\boldsymbol{m}_{ij}^{\ell+1} = \boldsymbol{m}_{ij}^{\ell} + \text{MLP}_{\boldsymbol{m}}(\text{BN}(\hat{\boldsymbol{m}}_{ij}^{\ell+1})) + \text{MLP}_t(\boldsymbol{t}) + \text{MLP}_f(\boldsymbol{f}),$$

$$\boldsymbol{h}_i^{\ell+1} = \boldsymbol{h}_i^{\ell} + \alpha(\text{BN}(U^{\ell}\boldsymbol{h}_i^{\ell} + \mathcal{A}_{j \in \mathcal{N}_i}(\sigma(\hat{\boldsymbol{m}}_{ij}^{\ell+1}) \odot V^{\ell}\boldsymbol{h}_j^{\ell}))),$$

where in layer $\ell$, $U^{\ell}, V^{\ell}, P^{\ell}, Q^{\ell}, R^{\ell} \in \mathbb{R}^{d \times d}$ and $\text{MLP}(\cdot)$ are learnable. $\text{MLP}(\cdot)$ with subscripts $\boldsymbol{m}, t, f$ all denote a 2-layer multilayer perceptron. $\alpha$, BN, $\mathcal{A}$, $\sigma$ denote the ReLU (Krizhevsky et al., 2010) activation, batch normalization (Ioffe, 2015), aggregation function SUM pooling (Xu et al., 2018) and sigmoid function, respectively. $\odot$ is the Hadamard product, $\mathcal{N}_i$ denotes the neighborhoods of node $i$. We use 12 layers with hidden dimension $d = 256$ following Sun & Yang (2023). Lastly, a Sigmoid activation is applied to the final layer embeddings of nodes or edges, which is then to predict a binary cross entropy loss between candidate solution graph vs. input solution graph.

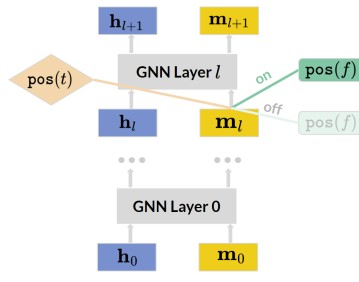

Figure 4: Architecture of objective-directed denoiser.

**Double Graph Conditioning.** It is worth noticing that our denoiser $\phi_{\theta}(\mathcal{G}_t^{\boldsymbol{x}}, \mathcal{G}, f(\mathbf{x}), t)$ takes two graphs as input: the problem graph $\mathcal{G}$ and the noisy solution graph $\mathcal{G}_t^{\boldsymbol{x}}$. To pass both graphs into the anisotropic GNN above, we concatenate the positional encoding of node/edge features in both graphs: suppose $\mathcal{G} = \langle \boldsymbol{V}, \boldsymbol{E} \rangle$ and $\mathcal{G}_t^{\boldsymbol{x}} = \langle \boldsymbol{V}_t^{\boldsymbol{x}}, \boldsymbol{E}_t^{\boldsymbol{x}} \rangle$, we pass $\boldsymbol{h}_i^0 = (\text{pos}(\boldsymbol{V}_{(i)}), \text{pos}(\boldsymbol{V}_{t,(i)}^{\boldsymbol{x}}))$ and $\boldsymbol{m}_{ij}^0 = (\text{pos}(\boldsymbol{E}_{(i,j)}), \text{pos}(\boldsymbol{E}_{t,(i,j)}^{\boldsymbol{

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

## A RIGOROUS DEFINITIONS OF CO PROBLEMS: TSP AND MIS

### A.1 DEFINITION OF TSP

**Problem 1** (Travelling Salesman Problem **(TSP)**). *TSP is defined on a fully connected undirected graph $\mathcal{G}$, where node feature $\boldsymbol{v}_i \in \mathbb{R}^2$ is the 2D coordinate of node $i$ and edge feature $\boldsymbol{e}_{ij}$ is the Euclidean distance between node $i$ and $j$. $\boldsymbol{e}_{ij} = \boldsymbol{e}_{ji}$ given $\mathcal{G}$ is undirected. The goal of TSP is to find the tour $\boldsymbol{x}$ covering all nodes that minimizes the total travelling distance. Thus, the constraints $c_i$'s in (1) require $\boldsymbol{x}$ to be a permutation over all nodes in $\mathcal{G}$; and the objective $f(\boldsymbol{x} \mid \mathcal{G})$ to minimize in (1) is defined by formula $\sum_{i=1}^{n-1} \boldsymbol{e}_{\boldsymbol{x}(i),\boldsymbol{x}(i+1)} + \boldsymbol{e}_{\boldsymbol{x}(n),\boldsymbol{x}(1)}$, counting total travelling distance in $\boldsymbol{x}$.*

### A.2 DEFINITION OF MIS

**Problem 2** (Maximum Independent Set **(MIS)**). *MIS is to find the largest independent set for any given undirected graph $\mathcal{G}$. An independent set of $\mathcal{G}$ is defined as a subset of its nodes where nodes are pair-wisely disconnected. We consider both unweighted and weighted versions of MIS: for weighted MIS, $\boldsymbol{v}_i \in \mathbb{N}^*$ is an integer recording the weight of node $i$; for unweighted case, $\boldsymbol{v}_i = 1$ for all nodes. Edges in $\mathcal{G}$ are binary: $\boldsymbol{e}_{ij} = 1$ means node $i$ and $j$ are connected otherwise disconnect. The $c_i$'s in (1) constraint that $\boldsymbol{e}_{x(i),x(j)} = 0$ are satisfied for all node pairs in $x$; and $f(\boldsymbol{x} \mid \mathcal{G})$ in (1) is defined as $\sum_{\boldsymbol{x}(i) \in \boldsymbol{x}} \boldsymbol{v}_{\boldsymbol{x}(i)}$, counting the (weighted) size of $\boldsymbol{x}$.*

## B OBJECTIVE-GUIDED DIFFUSION MODEL

### B.1 DETAILED FORWARD PROCESS

Following the forward process of discrete diffusion models (Vignac et al., 2022; Igashov et al., 2024), categorical noise is progressively add to the clean data $\boldsymbol{E}^{\boldsymbol{x}}$ sampled from $P(\boldsymbol{E}^{\boldsymbol{x}} \mid \mathcal{G})$ in the training dataset by formula (3), generating a sequence of latents $\boldsymbol{E}_0^{\boldsymbol{x}} := \boldsymbol{E}^{\boldsymbol{x}}, \boldsymbol{E}_{1:T}^{\boldsymbol{x}} := \boldsymbol{E}_1^{\boldsymbol{x}}, \boldsymbol{E}_2^{\boldsymbol{x}}, \cdots, \boldsymbol{E}_T^{\boldsymbol{x}}$:

$$q\left(\boldsymbol{E}_t^{\boldsymbol{x}} \mid \boldsymbol{E}_{t-1}^{\boldsymbol{x}}\right) = \mathrm{Cat}\left(\boldsymbol{E}_t^{\boldsymbol{x}}; p = \boldsymbol{E}_{t-1}^{\boldsymbol{x}} \boldsymbol{Q}_t\right) \quad \text{and} \quad q\left(\boldsymbol{E}_t^{\boldsymbol{x}} \mid \boldsymbol{E}^{\boldsymbol{x}}\right) = \mathrm{Cat}\left(\boldsymbol{E}_t^{\boldsymbol{x}}; p = \boldsymbol{E}^{\boldsymbol{x}} \overline{\boldsymbol{Q}}_t\right),$$
(3 recall)

where $\mathrm{Cat}(\cdot, p)$ denotes categorical distribution, $\boldsymbol{Q}_t$'s are transition kernel for categorical variables and $\overline{\boldsymbol{Q}}_t = \boldsymbol{Q}_1 \ldots \boldsymbol{Q}_t$. In (3), $\boldsymbol{E}^{\boldsymbol{x}}$ denotes the edge features in the solution graph and each $\boldsymbol{E}_t^{\boldsymbol{x}}$ (including $\boldsymbol{E}^{\boldsymbol{x}}$) is organized as an $n \times n \times d_e$ tensor with entries being one-hot vectors of $d_e$ categories. In the case where edge features are binary, $d_e = 2$ and $[\boldsymbol{Q}_t]_{i,j} = q\left(\boldsymbol{E}_t^{\boldsymbol{x}} = j \mid \boldsymbol{E}_{t-1}^{\boldsymbol{x}} = i\right)$ for $i, j \in \{0, 1\}$. A common choice of $\boldsymbol{Q}_t$ is $\alpha^t \begin{bmatrix} 1 & 0 \\ 0 & 1 \end{bmatrix} + (1 - \alpha^t) \begin{bmatrix} \frac{1}{2} & \frac{1}{2} \\ \frac{1}{2} & \frac{1}{2} \end{bmatrix}$ with cosine scheduling $\{\alpha^t\}$, transitioning the data distribution to the uniform one. In backward process, for binary case, each entry in $\boldsymbol{E}_T$ is sampled as the one-hot vector of a Bernoulli variable with probability $\frac{1}{2}$.

### B.2 DERIVATION FOR GUIDANCE

Ideally, we want to generate from $P(\mathcal{G}^{\boldsymbol{x}} \mid \mathcal{G}, f^*)$ with $f^*$ being the optimal objective for problem $\mathcal{G}$. Diffusion model approximates this distribution by iteratively sampling from $P(\mathcal{G}_{t-1}^{\boldsymbol{x}} \mid \mathcal{G}_t^{\boldsymbol{x}}, \mathcal{G}, f^*)$ for discrete time steps $t \in [T]$, to achieve this, $P(\mathcal{G}_0^{\boldsymbol{x}} \mid \mathcal{G}_t^{\boldsymbol{x}}, \mathcal{G}, f^*)$ is the key quantity to learn by diffusion model as demonstrated in (5). $P(\mathcal{G}_0^{\boldsymbol{x}} \mid \mathcal{G}_t^{\boldsymbol{x}}, \mathcal{G}, f^*)$ contains the information about objective value and Bayesian rule suggests:

$$P(\mathcal{G}_0^{\boldsymbol{x}} \mid \mathcal{G}_t^{\boldsymbol{x}}, \mathcal{G}, f^*) \propto P(\mathcal{G}_0^{\boldsymbol{x}} \mid \mathcal{G}_t^{\boldsymbol{x}}, \mathcal{G}) \cdot P(f^* \mid \mathcal{G}_t^{\boldsymbol{x}}, \mathcal{G}),$$
(6 recall)

where $P(\mathcal{G}_0^{\boldsymbol{x}} \mid \mathcal{G}_t^{\boldsymbol{x}}, \mathcal{G})$ on the RHS is the unconditioned probability to denoise $\mathcal{G}_t^{\boldsymbol{x}}$. (6) shows the difference between conditioned and unconditioned denoising probability is $P(f^* \mid \mathcal{G}_t^{\boldsymbol{x}}, \mathcal{G})$. Thus, $P(f^* \mid \mathcal{G}_t^{\boldsymbol{x}}, \mathcal{G})$ suggests the the direction to improve the optimality of generated solutions and quantitatively it equals to $\frac{P(\mathcal{G}_{t-1}^{\boldsymbol{x}} \mid \mathcal{G}_t^{\boldsymbol{x}}, \mathcal{G}, f^*)}{P(\mathcal{G}_{t-1}^{\boldsymbol{x}} \mid \mathcal{G}_t^{\boldsymbol{x}}, \mathcal{G})}$, which can be easily seen by dividing the denominator from (6) on both sides. Given that Alg.1 already approximates $P(\mathcal{G}_{t-1}^{\boldsymbol{x}} \mid \mathcal{G}_t^{\boldsymbol{x}}, \mathcal{G}, f^*)$, this property provides a way to further enhance the performance of Alg.1 by denosing with the following probability at each

step:

$$\frac{1}{Z} \cdot P(\mathcal{G}_0^{\boldsymbol{x}} \mid \mathcal{G}_t^{\boldsymbol{x}}, \mathcal{G}, f^*) \left( \frac{P(\mathcal{G}_0^{\boldsymbol{x}} \mid \mathcal{G}_t^{\boldsymbol{x}}, \mathcal{G}, f^*)}{P(\mathcal{G}_0^{\boldsymbol{x}} \mid \mathcal{G}_t^{\boldsymbol{x}}, \mathcal{G})} \right)^{\gamma}. \tag{7 recall}$$

In (7), $Z$ is a normalizing factor and $\gamma \geq 0$ controls the strength of guidance.

## C    INPUT GRAPHS CONCATENATION IN GNN

**TSP.** In TSP, the problem graph $\mathcal{G} = \langle \boldsymbol{V}, \boldsymbol{E} \rangle$ contains both node and edge features, where node feature is the node coordinate on 2D and edge is the distance between a pair of nodes (Problem 1). As another input to GNN, the noisy solution graph $\mathcal{G}_t^{\boldsymbol{x}} = \langle \boldsymbol{V}, \boldsymbol{E}_t^{\boldsymbol{x}} \rangle$ has only the edges to predict (§ 3.1), so we concatenate the edge features in both graph as input: by setting $\boldsymbol{m}_{ij}^0 = (\text{pos}(\boldsymbol{E}_{(i,j)}), \text{pos}(\boldsymbol{E}_{t,(i,j)}^{\boldsymbol{x}}))$, here the dimension of sinusoidal embedding for both $\boldsymbol{E}$ and $\boldsymbol{E}_t^{\boldsymbol{x}}$ is 128, half of the model hidden dimension. $\boldsymbol{h}_i^0$ is initialized as $\text{pos}(\boldsymbol{V}_{(i)})$. Given the distance information in $\boldsymbol{E}$ is already encoded in node coordinates $\boldsymbol{v}_i$, in experiments we observe that concatenating $\boldsymbol{E}$ into input edge embedding does not have significant advantage on model performance, compare to set $\boldsymbol{m}_{ij}^0 = \text{pos}(\boldsymbol{E}_{t,(i,j)}^{\boldsymbol{x}})$.

**MIS.** In a MIS problem $\mathcal{G} = \langle \boldsymbol{V}, \boldsymbol{E} \rangle$, $\boldsymbol{V}$ records the weights of nodes and $\boldsymbol{E}$ indicates the existence of edges (Problem 2); and the noisy solution graph $\mathcal{G}_t^{\boldsymbol{x}}$ has only the nodes to predict (§ 3.1). So we concatenate the node features in both graph as input by setting $\boldsymbol{h}_i^0 = (\text{pos}(\boldsymbol{V}_{(i)}), \text{pos}(\boldsymbol{V}_{t,(i)}^{\boldsymbol{x}}))$. The dimension of both $\text{pos}(\boldsymbol{V}_{(i)})$ and $\text{pos}(\boldsymbol{V}_{t,(i)}^{\boldsymbol{x}})$ is 128. $\boldsymbol{m}_{ij}^0$ is initialized as $\text{pos}(\boldsymbol{E})$. Unlike the case in TSP, concatenating $\boldsymbol{V}$ into input node embedding is essential for solving weighted MIS.

## D    ADDITIONAL TRAINING DETAILS

**Hardware.** Models are trained with NVIDIA A100 GPUs or H100 GPUs. Models for TSP-50 (with 12800/76800 training instances), TSP-100 (12800 training instances) and MIS-100 are trained with one GPU, models in other cases are trained with 4 GPUs with data parallelism.

**Training Details.** All GuideCO and DIFUSCO models are trained with a cosine learning rate schedule starting from $2e^{-4}$ and ending at 0.

- TSP-50: We test using 12800, 76800, 1502000 random instances labelled by heuristics to train GuideCO and DIFUSCO models, for 50 epochs.
  Experiments with 12800 instances are trained on 1xA100 GPU with batch size 128;
  Experiments with 76800 instances are trained on 1xA100 GPU with batch size 128;
  Experiments with 1502000 instances are trained on 4xH100 GPUs with effective batch size 1024.

- TSP-100: We use 12800 76800 or 1502000 random instances labelled by heuristics to train GuideCO and DIFUSCO models, for 50 epochs. Training 12800 is with batch size 64, in other cases, batch size is 256.

- TSP-500: We use 128000 random instances labelled by heuristics to train GuideCO and DIFUSCO models, for 50 epochs with a batch size of 64.

- TSP-1000: We use 64000 random instances labelled by heuristics to train GuideCO and DIFUSCO models, for 50 epochs with a batch size of 16.

- MIS-100 (Weighted): We use 12800 randomly sampled training instance and train GuideCO and DIFUSCO for 50 epochs with a batch size of 32. Tested on one A100 GPU.

- SATLIB: We use the training split of 39500 examples from [46, 92] and train GuideCO and DIFUSCO for 50 epochs with a batch size of 64. Tested on four A100 GPUs.

- ER-[700-800]: We use 163840 random instances with heuristic lables and train GuideCO and DIFUSCO for 50 epochs with a batch size of 16. Tested on four H100 GPUs.

# E  ADDITIONAL PLOTS

An additional interesting observation in TSP-50 experiment is that the best guidance strength level is shifting left and the improvement of guidance is fading as the number of training instances increases.

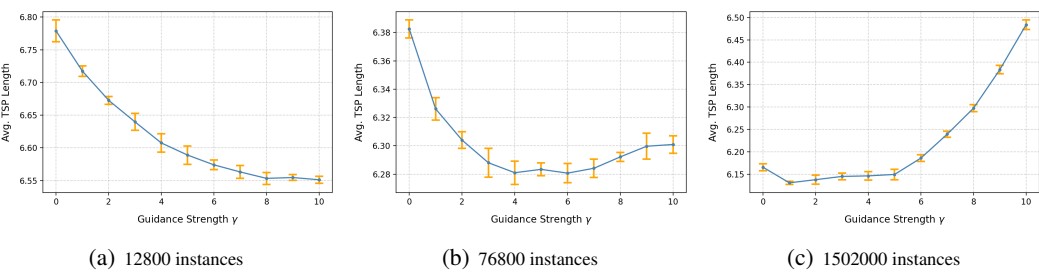

(a) 12800 instances   (b) 76800 instances   (c) 1502000 instances

Figure 8: **Guidance effect curve under varying the number of training instances.** Reported on TSP-50.