# OpenReview forum: "GuideCO: Training Objective-Guided Diffusion Solver with Imperfect Data for Combinatorial Optimization"
_ICLR.cc/2025/Conference — ICLR 2025 Conference Withdrawn Submission_

### Official Review · Reviewer_rPmF · 2024-10-28

**Soundness:** 2
**Presentation:** 2
**Contribution:** 2
**Rating:** 3
**Confidence:** 5

**Summary:**

The authors proposed GuideCO, a method that uses diffusion models to solve two CO problems: MIS and TSP. The authors proposed a training and inference algorithm with guidance that encode the problem instance information. Furthermore, GuideCO maps the the output of the diffusion model at time 0 (logits) using an adopted greedy decoding procedure.

**Strengths:**

- The paper is generally well-written and easy to follow.

- The observation of the exponential scaling law between the optimality gap and the amount of training data needed for training DM-based CO solvers.

- Evaluation on on two distinct and challenging CO problems: MIS and TSP.

- Employing the "Objective-Aware Graph-Based Denoiser" architecture for CO problems.

**Weaknesses:**

- Recognizing that DIFUSCO (or any other NN-based supervised CO solver) as "trained" with optimally labeled data is not accurate. For example, DIFUSCO uses KaMIS (or ReduMIS [6]) to label their graphs. This method is a heuristic that does not guarantee an optimal MIS. The optimality and perfectly labelled claims here need to be revisited. This is a major issue with the paper as the authors develop their method based on this observation. For the MIS problem with ER graphs, the only difference between how DIFUSCO and GuideCO label graphs is using a different hueristic. DIFUSCO uses KaMIS whereas GuideCO uses a faster matlab function that cites [11].

- The link between the proposed framework and their bi-level optimization formulation is weak. It is not wrong with the (not very justified) approximation assumption of the lower level problem, but I don't see the benefit. Further clarification is needed here in addition to the "approximation" assumed for solving the lower level problem. To me, the lower level problem is just ensuring that the solution is a valid one (within the feasible set of the original problem constraints).

- Missing OOD generalization evaluation: DIFUSCO is known to have weak generalization at inference time (see Table 5 in [8] for an example). Can the authors report similar results to evaluate the generalization of the proposed solver?

- Missing several baselines for comparisons such as [2,4,8,10,13].

**Questions:**

### Major Comments/Questions:

- The amount of data in training is not a challenge if the proposed solver can generalize and is capable of performing well when trained on imperfectly labelled graphs as we can always use graph random generators to generate training graphs.

- In lines 48 and 48, the authors say that DIFUSCO had achieved SOTA results which is not entirely true as DIFUSCO's best configuration achieved on par results with CONCORDE.

- The simplicity of training DMs for CO problems is not supported. How is it simple? It is very time-consuming. Further discussion is needed here.

- The paper is missing discussions on unsupervised data-centric (such as [1]) and data-less solvers (such as [2] and [3]). This motivates the question: Why do we need supervised learning when unsupervised methods are performing well?

- DIFUSCO is no longer the SOTA supervised DM CO solver. T2T in [4] is. Furthermore, consistency models have also been recently employed in [5].

- Results to understand the impact of the solution generated by the diffusion model (first step) is needed. For example, if we initialize the greedy decoding procedure with any Maximal independent set (or their logits), do we still see the same results? If not, then how much improvement do we gain from the solution provided by the DM?

- The proposed method aims at merging a DM-based solution (or its logits) with heuristics. If the combined method (i.e., GuideCO) still can **not** outperform a problem-specific heuristics solver (in this case KaMIS or Concorde), what is the benefit of using GuideCO?

- For Table 5, why Gurobi is not reported for ER? Gurobi achieves an average MIS size of 41.38 in total runtime of 50 minutes from DIFUSCO's Table 2. Why there are no ER results for the last two rows? Furthermore, it seems that two different sets of GPUs were used here. Additionally, KaMIS has a heuristic solver specifically for the weighted MIS problem (See [12]).

- How scalable is the proposed solver? For example, can GuideCO solve the TSP instances in [9]? Or the larger graphs reported in Table 3 of LwD [10]?

- Run-time (in this case inference run-time) results are not reported for the TSP problem (Tables 2, 3, and 4).

### Minor Comments/Questions:

- Is there reason why GuideCO with $\gamma=0$ not reported in the MIS table (like was done for the TSP tables)?

- How sensitive is training the DM to the value of $p$?

- I is undefined in (1).

- To seek in line 181.

- Missing comas in line 212. It should be i = 1,...,n-1.

- "In pracThe" in line 227

- In line 248, is $E^x_{1:T}$ a set or a sequence?

- Shouldn't there be a transpose for $\bar{Q}_{t-1}$ in (5)? Also, $\odot$ is not defined until the subsequent subsubsection.


### References:

[1] UNSUPERVISED LEARNING FOR COMBINATORIAL OPTIMIZATION NEEDS META LEARNING

[2] Combinatorial Optimization with Physics-Inspired Graph Neural Networks

[3] A differentiable approach to the maximum independent set problem using dataless neural networks

[4] T2T: From Distribution Learning in Training to Gradient Search in Testing for Combinatorial Optimization

[5] OptCM: The Optimization Consistency Models for Solving Combinatorial Problems in Few Shots

[6] Finding near-optimal independent sets at scale

[7] Fast local search for the maximum independent set problem

[8] Dataless Quadratic Neural Networks for the Maximum Independent Set Problem

[9] Solving the Large-Scale TSP Problem in 1 h: Santa Claus Challenge 2020

[10] Learning What to Defer for Maximum Independent Sets

[11] A simple algorithm to optimize maximum independent set (Advanced Modeling and Optimization 2010).

[12] Finding Near-Optimal Weight Independent Sets at Scale (https://arxiv.org/pdf/2208.13645)

[13] Google, Inc. Google or-tools. 2022. URL https://developers.google.com/optimization.

---

### Official Review · Reviewer_HeiH · 2024-11-01

**Soundness:** 3
**Presentation:** 3
**Contribution:** 3
**Rating:** 3
**Confidence:** 4

**Summary:**

This paper presents a novel approach, GuideCO, designed to address challenges in training diffusion models for combinatorial optimization problems. Diffusion models often require large amounts of high-quality, near-optimal training data to perform well, which can be costly and difficult to obtain. GuideCO mitigates this by enabling effective training on imperfect or sub-optimal data through an objective-guided diffusion model enhanced by classifier-free guidance. This two-stage generate-then-decode framework allows for high-quality solution generation across tasks like the TSP and MIS, demonstrating superior performance to existing diffusion solvers, particularly in data-scarce scenarios. Experimental results show GuideCO’s ability to achieve competitive or better performance even when trained with less optimal data, offering a scalable solution for CO problems where perfect data is not accessible.

**Strengths:**

This paper established a new framework for solving combinatorial optimization problems when the label quality is low. This topic is interesting as low quality labels are easier to obtain. The proposed solver performs better than DIFUSCO, demonstrating its effectiveness in this scenario.

**Weaknesses:**

1.	The decode process should be elaborated in detail. Besides, are there any reasons for selecting greedy methods? It seems that MCTS works better in most cases for TSP.
2.	The experimental results are weak. More comparisons with other CO methods on larger datasets are needed. Besides, runtime comparisons are also needed.
3.	I cannot find practical meanings of the proposed method. On TSP-50, the optimal solutions can be obtained easily, but GuideCO still has over 10% gaps. On TSP-1000, the gap is even 37%, which is totally meaningless.

**Questions:**

1.	What is the difference between Q1 and Q2 in Introduction?
2.	The value of f_tar is set as the average optimal objective value in validation set. But you under the scenario when labels are obtained by heuristic and there is no optimal. Where does the optimal objective value come from?
3.	On such small scale graphs, why 12 layers of GNN are needed?
4.	The citations seem to be incorrect. There are no hyperlinks to the corresponding references.

---

### Official Review · Reviewer_vEQu · 2024-11-03

**Soundness:** 3
**Presentation:** 2
**Contribution:** 1
**Rating:** 5
**Confidence:** 4

**Summary:**

This paper introduces GuideCO, an objective-guided diffusion solver designed to address limitations in existing diffusion models for combinatorial optimization (CO) tasks. While recent diffusion-based generative models can generate near-optimal solutions for CO problems like the Traveling Salesman Problem (TSP) and Maximum Independent Set (MIS), they typically require large amounts of high-quality, optimally-labeled training data. GuideCO overcomes this limitation by employing a two-stage generate-then-decode framework with an objective-guided diffusion model, which utilizes classifier-free guidance to produce high-quality solutions even with imperfectly labeled data. Experimental results demonstrate the effectiveness of the proposed method.

**Strengths:**

1. Introducing objective guidance through the conditional training of generative models is an interesting approach that is intuitively expected to enhance effectiveness.

2. The paper is easy to follow.

**Weaknesses:**

1. Some design elements lack novelty. The generate-then-decode framework has long been established as a default pipeline for non-autoregressive neural solvers, e.g. [1] [2] [3], which first generate a solution heatmap and then decode a feasible solution from that heatmap.

2. The main methodological proposal—introducing objective guidance through the conditional training of generative models—has already been explored for diffusion models in other related contexts [4][5]. This diminishes the technical contribution of the paper, particularly given that the empirical results do not adequately compare to previous state-of-the-art approaches.

3. The empirical results do not support the claim that this method can enhance the performance of state-of-the-art models. The authors claim that "Please note experiments are conducted for scenarios where training data is labeled by heuristics but not exact solvers." However, the authors still have to show evidence that the proposed method can improve upon the current advanced results in this field, especially when the proposed method can be directly applied to near-optimal supervision.

[1] DIMES: A Differentiable Meta Solver for Combinatorial Optimization Problems. NeurIPS 2022.

[2] DIFUSCO: Graph-based Diffusion Solvers for Combinatorial Optimization. NeurIPS 2023.

[3] T2T: From Distribution Learning in Training to Gradient Search in Testing for Combinatorial Optimization. NeurIPS 2023.

[4] Is Conditional Generative Modeling all you need for Decision Making? ICLR 2023.

[5] Diffusion model for data-driven black-box optimization. 2024.

**Questions:**

1. How is the proposed generate-then-decode framework different from previous methods inluding DIMES, DIFUSCO?

Please also see respond to the points listed in the weaknesses section.

---

### Official Review · Reviewer_zQTz · 2024-11-03

**Soundness:** 3
**Presentation:** 2
**Contribution:** 2
**Rating:** 3
**Confidence:** 4

**Summary:**

The paper  introduces GuideCO, a diffusion-based framework designed to solve combinatorial optimization (CO) problems efficiently, with imperfectly labeled data. Traditional diffusion solvers for CO require high-quality, exact or near-optimal solutions to perform well, which can be costly and challenging to obtain. GuideCO addresses this by introducing a two-stage generate-then-decode framework that guides solution generation using sub-optimal data.

**Strengths:**

The exponetial scaling law is interesting. The authors identify a scaling relationship between optimality gap and data quantity. GuideCO attempts to bridge this gap by introducing objective-guided generation.

**Weaknesses:**

1.The technical contribution is limited. The guidance that the proposed method adopts on diffusion models is similar with diffusion model as plug-and-play priors [1]. Please correct me if I make any mistakes.

2.The empirical performance improvement is limited.

2.1In comparison with the baseline DIFUSCO:  solving small instances takes very cheap time for the exact solver, thus we could assume it's actionable to directly generate huge amount of training data labeled with the exact optimal solution. In this case, there remains a huge gap between the DIFUSCO+exact and GuideCO when one has enough training data.
Moreover, when solving larger instances, instead of training DIFUSCO with the exact optimal solution, LKH-3 is also a much powerful heuristic to label the training data with the size of 500, 1000, 10000[2]. According to Table4, on large instances there is still a gap between GuideCO and DIFUSCO. When one wants to train a solver that could solve real-world problems, using LKH-3 with DIFUSCO seems still to be the first choice. It would be beneficial to show the superiority of GuideCO over DIFUSCO when labeling the optimal solution is impractical, (e.g. on TSPs with the size of at least 10,000) .

2.2 It would be interesting to investigate: Can GuideCO further improve the performance when trained on exact labels?

2.3 (Following 2.1) Though trained on imperfect labels, the paper still does not consider scalability issues, the method only conducts experiments up to 1000 nodes. If GuideCO shows superiority over existing baselines on large scale data where labeling the exact labels are impractical, the proposed method would then significantly illustrate its potential for practical application.

[1] Diffusion models as plug-and-play priors. Alexandros Graikos, Nikolay Malkin, Nebojsa Jojic, Dimitris Samaras. Neurips 2022.

[2]DIFUSCO: Graph-based Diffusion Solvers for Combinatorial Optimization. Zhiqing Sun, Yiming Yang. Neurips 2023.

**Questions:**

There is a typo in line 227.

Please refer to the weakness.

---

### Note · Authors · 2024-12-02

I have read and agree with the venue's withdrawal policy on behalf of myself and my co-authors.